# Three Uses of One Neural Network: Automatic Segmentation of Kidney Tumor and Cysts Based on 3D U-Net

Yi Lv [1,2], Junchen Wang [2]

[1] North China Research Institute of Electro-optics, Beijing, 100015, China
[2] School of Mechanical Engineering and Automation, Beihang University, Beijing, 100191, China
dnkii@buaa.edu.cn

**Abstract.** Medical image processing plays an increasingly important role in clinical diagnosis and treatment. Using the results of kidney CT image segmentation for three-dimensional reconstruction is an intuitive and accurate method for diagnosis. In this paper, we propose a three-step automatic segmentation method for kidney, tumors and cysts, including roughly segmenting the kidney and tumor from low-resolution CT, locating each kidney and fine segmenting the kidney, and finally extracting the tumor and cyst from the segmented kidney. The results show that the average dice of our method for kidney, tumor and cysts is about 0.93,0.57,0.73.

**Keywords:** Medical Image Segmentation, Deep Learning, Neural Network

## 1 Introduction

Segmentation and reconstruction of CT or MRI medical images is the main source of navigation data, i.e. anatomical structure of tissues and organs[1-4]. At present, the most commonly used method of medical image segmentation is still manual segmentation, which takes a long time and depends on the operator's experience and skills[5]. In recent years, breakthroughs have been made in the research of neural networks[6-9]. The deep learning technology based on neural networks can achieve fast segmentation, and effectively solve the problem of low accuracy and long time-consuming image segmentation[10]. In the field of medical image segmentation, the breakthrough of in-depth learning began with the introduction of Full Convolutional Neural Network (FCN), and another breakthrough of neural network architecture U-Net made it possible to achieve high-precision automatic segmentation of medical images. Long Jonathan et al.[11] proposed Fully Convolutional Networks structure in 2015. In the same year, Olaf Ronneberger et al.[12] proposed the U-Net network structure. U-Net is a semantic segmentation network based on FCN, which is suitable for medical image segmentation. With the proposal of 3D convolutional neural networks such as 3D U-Net [13] and V-Net [14], the segmentation accuracy of some organs has reached a milestone. For example, in the MICCAI challenge 2019 kits19 competition, the accuracy of 3D U-Net in the task of kidney segmentation is very close to that of human, but the required time to complete a segmentation is far less than that of manual segmentation. The deep learning-based methods not only surpass the traditional algorithms, but also approach the accuracy of manual segmentation.

In this paper, we propose a three-step automatic segmentation method for kidney, tumors and cysts, including roughly segmenting the kidney and tumor from low-resolution CT, locating each kidney and fine segmenting the kidney, and finally extracting the tumor and cyst from the segmented kidney.

## 2 Methods

### 2.1 Network Architecture

The network architecture (3D U-Net) is illustrated in Fig. 1. We choose 3D U-Net as the neural network for the all three steps. 3D U-net includes an encoding path and a decoding path, each of which has four resolution levels. Each layer of the encoding path contains two $3 \times 3 \times 3$ convolution, each followed by a ReLu layer, followed by a $2 \times 2 \times$ Maximum pool layer with step size of 2 in each direction of 2. In the decoding path, each layer contains a 2 with a step size of $2 \times 2 \times 2$, followed by two $3 \times 3 \times 3$, each followed by a RuLu layer.

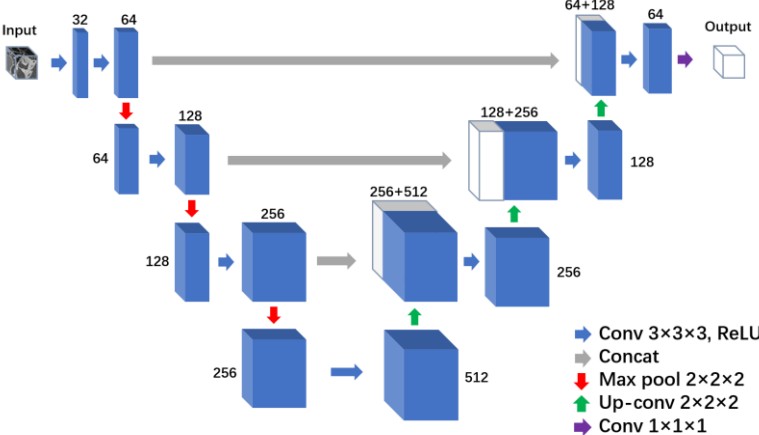

Figure 1. The network architecture. Blue cuboids represent feature maps. The number of channels is denoted next to the feature map.

## 2.2  Segmentation from Low-Resolution CT

Considering that the complete CT sequence is too large for the GPU, we scaled the spacing of all CT sequences to 3.41, 1.71 and 1.71. Then we cut the size of the sequence to 128 * 248 * 248 under the condition of constant spacing. Finally, we uniformly scale all the sequence sizes to 96 * 128 * 128 for the training of neural network. Then we use the generated data for training of the 3D U-Net for the first step.

## 2.3  Fine Segmentation of Kidney

According to the results of rough segmentation in the previous step, we first locate the position of two (or one) kidneys. Secondly, we crop the CT image with the region located and scale the size to 80 * 128 * 128. Then, we set the kidney, tumor and cyst to a unique label as the training data for the second step. Finally, we conduct the data augmentation including random translation, scaling, and trained the second 3D U-Net with the generated data.

## 2.4  Segmentation of Tumor and Cysts

To prevent the network from identifying areas outside the kidney as tumors or cysts, only the region of kidney segmented in the above step is used as the input for the network in step 3. Tumor and cysts were labeled with 1 and 2, respectively. After segmentation, we scale the segmented results to the original size as the final output.

## 2.5  Training protocols

All the algorithms were implemented using Pytorch1.2 with Python3.7 and ran on a workstation with a AMD 5800X CPU, 32G memory and a NVIDIA RTX8000 GPU. The training protocols of the proposed method is shown in Table 1.

Table I. Training protocols

| | |
|---|---|
| **Data augmentation methods** | Scaling, rotations, brightness, contrast, gamma |
| **Initialization of the network** | Kaiming normal initialization |
| **Batch size** | 4 |
| **Total epochs** | 50 |
| **Loss Function** | Dice loss and weighted cross entropy |

# 3 Results

As the accuracy metrics, the Dice similarity coefficient (DSC), average symmetric surface distance (ASSD [mm]) and surface distance deviation (SDD [mm]) between the predicted mask and the ground truth mask were employed. Assume A and B are two masks, these metrics are given by (1), (2) and (3), where S(A) and S(B) are the surface points of A and B, respectively, and d(a, S(B)) is the minimum Euclidian distance between the point a and the points on the surface S(B). We perform 5-fold cross-validation on 300 data sets, but due to limited time, only one-fold has finished at the time of submission. The result is shown in Table II. Figure 2 shows the results with voxel-based rendering from three examples in the evaluation dataset. Figure 3 shows the segmentation results on CT slices.

$$\text{DSC} = \frac{2(A \cap B)}{A+B} \tag{1}$$

$$\text{ASSD} = \frac{1}{|S(A)|+|S(B)|} \times \left( \sum_{a \in S(A)} d(a, S(B)) + \sum_{b \in S(B)} d(b, S(A)) \right) \tag{2}$$

$$\text{SDD} = \sqrt{\frac{1}{|S(A)|+|S(B)|} \times \left( \sum_{a \in S(A)} (d(a, S(B)) - \text{ASSD})^2 + \sum_{b \in S(B)} (d(b, S(A)) - \text{ASSD})^2 \right)} \tag{3}$$

Table II. Dice Comparison on Three Structures

|        | Dice | ASSD (mm) | SSD (mm) |
|--------|------|-----------|----------|
| Kidney | 0.93 | 1.24      | 2.45     |
| Tumor  | 0.57 | 8.24      | 10.86    |
| Cysts  | 0.73 | 6.70      | 6.90     |

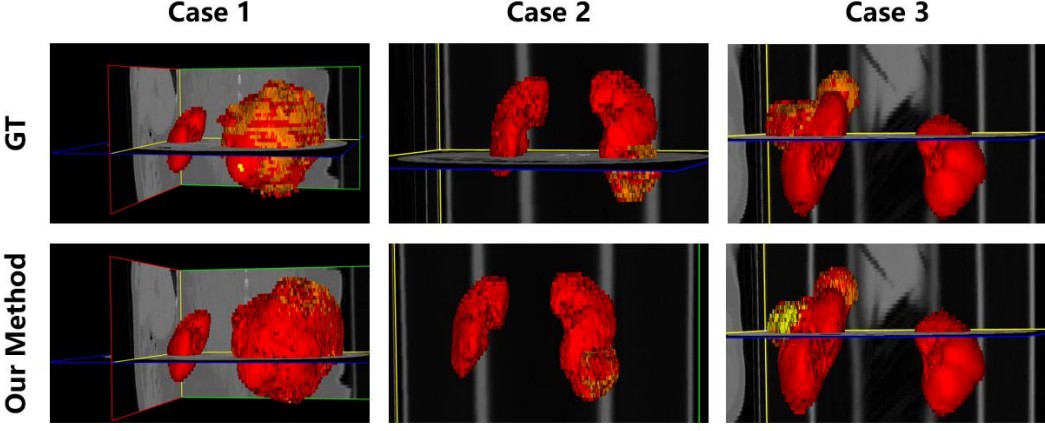

Figure 2. Segmentation results with voxel-based rendering from three examples in the evaluation dataset. For each example, the ground truth and the segmentation results are given for visual comparison. (yellow: cysts, red: kidney, brown: tumor)

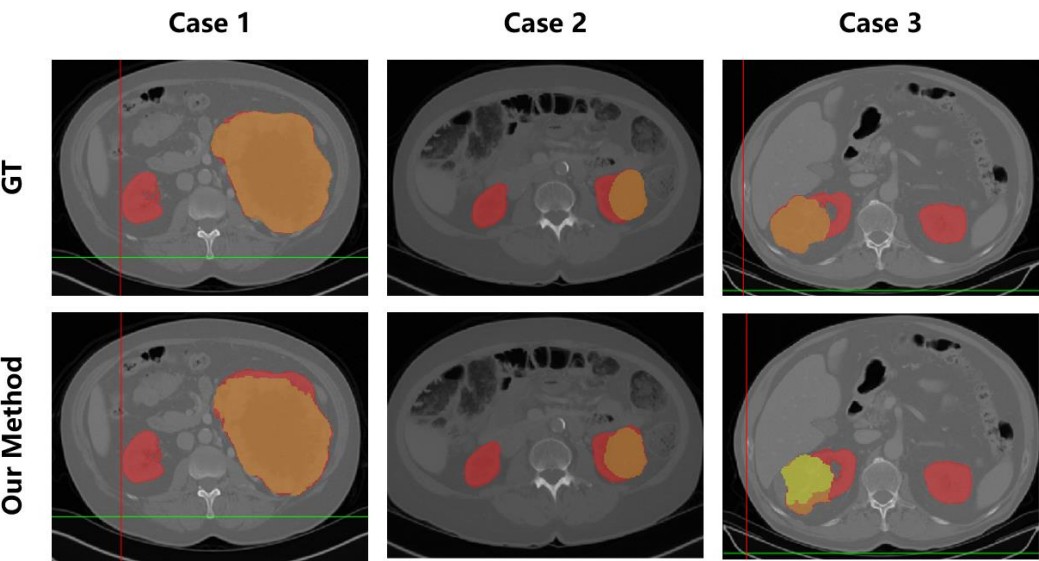

Figure 3. Segmentation results on CT slices. The two rows are the ground truth and segmentation results of our method. (yellow: cysts, red: kidney, brown: tumor)

## 4  Discussion and Conclusion

We propose a three-step automatic segmentation method for kidney, tumor and cysts based on 3D U-Net. The results show that the average dice of our method for kidney, tumor and cysts is about 0.93,0.57,0.73. The results show that the accuracy of our method on kidney is better than that of tumor and cysts. The region of the kidney can be accurately identified, but the accuracy of tumor and cysts is not satisfactory. As limited by the competition time, the neural network requires more time to be fully trained. Future work will focus on promoting accuracy of our method on the tumors and cysts.

## Acknowledgements

Junchen Wang was funded by National Key R&D Program of China (2017YFB1303004) and National Natural Science Foundation of China (61701014, 61911540075).

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
