# OpenReview forum: "Three Uses of One Neural Network: Automatic Segmentation of Kidney Tumor and Cysts Based on 3D U-Net"
_MICCAI.org/2021/Challenge/KiTS — Submitted to KiTS21 Challenge_

### Official Review · Reviewer_UMmB · 2021-08-30

**Rating:** 5

**Review:**

If I'm understanding correctly, the authors used a single network architecture for three different stages of a coarse-to-fine approach in which the kidney is first roughly localized, then finely segmented, then the tumors and cysts and simultaneously segmented from only the finely segmented kidney region. The paper contains several grammatical errors and is unfortunately brief. The authors should aim to add much greater detail in their revision. The KiTS21 paper template can be consulted for a list of content that the authors should aim to include/address.

---

### Official Review · Reviewer_cDP7 · 2021-08-30

**Rating:** 6

**Review:**

### Overall

- An institutional email address is preferred over foxmail if possible

### Introduction

- segmen-tation is hyphenated unnecessarily

### Methods

- "2 x two" should consistently either use 2 or two
- si ze has an erroneous space in it
- "3 x three" should consistently either use 3 or three

### Results

- Please expand this section considerably. Once offical results are known you can add these. It is also nice to add a figure that shows one or more predictions in comparison with the ground truth. Did you experiment with any other approaches that you decided not to use? What were the performance metrics for those? Why did you decide not to use them?

### Discussion and Conclusion

- Please also expand this section. What did you learn about the problem? Why do you feel your approach performed the way that it did? Did you notice any error modes in your model's predictions?

---

### Decision · Program_Chairs · 2021-08-30

**Decision:**

Major Revisions

**Comment:**

Please address the reviewer comments and resubmit